# Mustard Seed (*Brassica nigra*) Extract Exhibits Antiproliferative Effect against Human Lung Cancer Cells through Differential Regulation of Apoptosis, Cell Cycle, Migration, and Invasion

**DOI:** 10.3390/molecules25092069

**Published:** 2020-04-29

**Authors:** Asmaa Gamal Ahmed, Usama Khamis Hussein, Amr E. Ahmed, Kyoung Min Kim, Hamada M. Mahmoud, Ola Hammouda, Kyu Yun Jang, Anupam Bishayee

**Affiliations:** 1Department of Pathology, Chonbuk National University Medical School, Jeonju 54907, Korea; asmaascience3@gmail.com (A.G.A.); usamahussein@jbnu.ac.kr (U.K.H.); kmkim@jbnu.ac.kr (K.M.K.); 2Department of Biotechnology and Life Sciences, Faculty of Postgraduate Studies for Advanced Sciences, Beni-Suef University, Beni-Suef 62511, Egypt; amreahmed@psas.bsu.edu.eg; 3Research Institute of Clinical Medicine of Jeonbuk National University-Biomedical Research Institute of Chonbuk National University Hospital, Jeonju 54907, Korea; 4Faculty of Science, Beni-Suef University, Beni-Suef 62511, Egypt; mhmada@aucegypt.edu (H.M.M.); ola.hammouda@gmail.com (O.H.); 5Lake Erie College of Osteopathic Medicine, Bradenton, FL 34211, USA

**Keywords:** *Brassica nigra*, lung cancer, proliferation, DNA damage, apoptosis

## Abstract

Lung cancer is the primary cause of cancer-related death worldwide, and development of novel lung cancer preventive and therapeutic agents are urgently needed. *Brassica nigra* (black mustard) seeds are commonly consumed in several Asian and African countries. Mustard seeds previously exhibited significant anticancer activities against several cancer types. In the present study, we have investigated various cellular and molecular mechanisms of anticancer effects of an ethanolic extract of *B. nigra* seeds against A549 and H1299 human non-small cell lung cancer cell lines. *B. nigra* extract showed a substantial growth-inhibitory effect as it reduced the viability and clonogenic survival of A549 and H1299 cells in a concentration-dependent manner. *B. nigra* extract induced cellular apoptosis in a time- and concentration-dependent fashion as evidenced from increased caspase-3 activity. Furthermore, treatment of both A549 and H1299 cells with *B. nigra* extract alone or in combination with camptothecin induced DNA double-strand breaks as evidenced by upregulation of γH2A histone family member X, Fanconi anemia group D2 protein, Fanconi anemia group J protein, ataxia-telangiectesia mutated and Rad3-related protein. Based on cell cycle analysis, *B. nigra* extract significantly arrested A549 and H1299 cells at S and G2/M phases. Additionally, *B. nigra* extract suppressed the migratory and invasive properties of both cell lines, downregulated the expression of matrix metalloproteinase-2 (MMP2), MMP9, and Snail and upregulated the expression of E-cadherin at mRNA and protein levels. Taken together, these findings indicate that *B. nigra* seed extract may have an important anticancer potential against human lung cancer which could be mediated through simultaneous and differential regulation of proliferation, apoptosis, DNA damage, cell cycle, migration, and invasion.

## 1. Introduction

Lung cancer represents one of the most leading causes of cancer-related death worldwide with up to 1.9 million deaths occurring per year [1]. About 234,030 new cases of lung cancer and 154,050 deaths have been estimated to occur in the United States during 2018 [2]. Histopathologically, lung cancer is categorized into two main groups, namely small-cell lung cancer (15% of all lung cancers) and non-small cell lung cancer (NSCLC, 85% of all lung cancers). NSCLCs include diverse types of carcinomas, such as large-cell carcinomas, squamous cell carcinomas, and adenocarcinomas [3]. The leading causes of NSCLC include smoking and daily exposure to various environmental carcinogens [2,4,5]. Despite the development of novel targeted therapies, the overall survival of patients with NSCLC is only 15%. Unfortunately, in some patients the long term-use of the targeted therapies leads to drug resistance and disease recurrence [6,7]. Hence, it is a high priority to develop advanced detection techniques and new therapeutic strategies for treatment and prevention of NSCLC in high-risk population [8].

Consumption of cruciferous vegetables seems to reduce the risks of different kinds of cancers, including colorectal, renal, and prostate cancers, primarily due to the presence of sulfur-containing compounds glucosinolates [9,10]. However, the cancer preventive evidence is not strong for oral, lung, and breast cancer [9]. The glucosinolates are hydrolyzed by the enzyme myrosinase present in the crucifers to yield biologically active compounds isothiocyanates. The extracts/fractions of various fresh cruciferous vegetables, including cauliflower, broccoli and Brussels sprouts, have been shown to possess antiproliferative and antioxidant activities against human colon cancer cells [11]. A combined treatment of phenethyl isothiocyanate (derived from glucosinolate named gluconasturtiin) with conventional anticancer drugs exhibited a synergistic antiproliferative effect on myeloma tumor cell lines [12].

*Brassica nigra* (L.) W.D.J.Koch (popularly known as black mustard, family Brassicaceae) is an annual erect herb cultivated in the Mediterranean region in addition to various South-East countries. This dietary plant has been used in the traditional medicines for the treatment of neuralgia spasms, alopecia, snakebite, epilepsy, toothache, and various carcinomas [13]. Mustard oil is known to stimulate hair growth, and the mustard flour is considered an effective antiseptic agent [14]. The seeds of *B. nigra* contain about 4% isothiocyanate (sinigrin and myrosin), and more than 90% of isothiocyanates allyl isothiocyanate. The mustard seeds also contain about 30% of proteins, 27% fixed oil, inosite, lecithin, albumins, and mucilage [15,16,17].

*Brassica* vegetables are known to possess cancer preventive and therapeutic potential against broad ranges of cancer types, such as ovary, colon, bladder, lung and breast [18,19,20,21]. The ethanolic, hexane and ethyl acetate extracts of *B. nigra* have been found to exhibit antiproliferative activities against human hepatocellular (HepG2), cervical (HeLa), colorectal (HCT), and breast carcinoma (MCF-7) cells [22]. An extract of *B. nigra* has been able to protect HepG2 cells against benzo[a]pyrene-induced DNA damage, possibly via mechanisms associated with induction of detoxification enzymes [23]. The cytotoxic effects of allyl isothiocyanate, present in mustard seeds, have been reported against lung [24] and bladder cancer cells [25,26]. Moreover, previous studies reported that dietary mustard seeds suppressed azoxymethane-induced colon adenomas in mice [27] and dimethylhydrazine-induced colorectal carcinomas in rats [28]. Allyl isothiocyanate inhibited the growth of Ehrlich ascites tumor in mice by proapoptotic and antiangiogenic mechanisms [29]. Sinigrin, a major phytochemical present in *B. nigra*, inhibited diethylnitrosamine-initiated hepatocarcinogenesis in rats [30].

The anticancer potential and possible mechanisms of action of *B. nigra* seeds in lung cancer have not been explored before. Therefore, in the present study, we have investigated the antiproliferative effect of *B. nigra* seed extract against two human non-small cell lung cancer cells, namely A549 and H1299 cells. A549 cells (human alveolar basal epithelial adenocarcinoma) are characterized by wild-type p53, while H1299 cells (human epithelial adenocarcinoma) have homozygous partial deletion of the *TP53* gene and mutant *NRAS* gene [31]. The effect of the *B. nigra* extract on apoptosis, cell cycle distribution and replication stress-associated DNA damage and repair have also been studied. Finally, we explored possible antimigratory and anti-invasive properties of *B. nigra* extract and associated gene expression.

## 2. Results

### 2.1. B. nigra Extract Exhibited Antiproliferative Activities against A549 and H1299 Cells

The chemotherapeutic potential of *B. nigra* extract has been assessed by treating A549 and H1299 cells with various concentrations of the extract followed by determination of the viability and cytotoxicity (Figure 1A). The half maximal inhibitory concentration (IC_50_) of *B. nigra* extract was determined after treatment of A549 and H1299 lung cancer cells for 72 h. The IC_50_ values are found to be 32.02 and 25.38 μg/mL against the A549 and H1299 cell lines, respectively (Figure 1B). The anchorage-dependent growth and clonogenic potential of both A549 (Figure 1C,D) and H1299 cells (Figure 1E,F) were significantly affected by *B. nigra* extract in a concentration-dependent manner. Accordingly, *B. nigra* extract showed substantial growth-inhibitory effect on A549 and H1299 cells.

### 2.2. B. nigra Extract Induced Apoptosis through Promotion of Caspase-3 Activity.

The kinetic activity of caspase-3 was increased in *B.* nigra-treated A549 (Figure 2A) and H1299 (Figure 2C) cells in a time-dependent manner. After 24 h exposure of both cell lines to *B. nigra* extract, the activity of caspase-3 was significantly increased at 3 h following the treatment of the substrate solution containing ASP-GLu and it was found to be 2-fold of the control. On the other hand, the expression of cleaved caspase-3 was upregulated in a time-dependent manner at the protein level in A549 (Figure 2B) and H1299 (Figure 2D) cells. These findings indicate that *B. nigra* extract has the ability to induce apoptosis of A549 and H1299 cells.

### 2.3. B. nigra Extract Induced Replication-Associated DNA Damage 

DNA damage induced by replication stress leads to collapsed replication forks, which stimulates ataxia telangiectesia mutated (ATM)- and Rad3-related protein (ATR)-mediated cell cycle checkpoint responses. Moreover, the expression of Rad18, Fanconi anemia group D2 protein (FANCD2) and Fanconi anemia group J protein (FANCJ) proteins were also upregulated [32]. To investigate whether *B. nigra* extract induces replication stress-associated DNA damage and repair (DDR), A549 and H1299 cells were exposed to 30 and 20 μg/mL of *B. nigra* extract, respectively, alone or in combination with 500 nM camptothecin (CPT). The whole protein lysates from the cells were probed with various antibodies for DNA damage and repair proteins. *B. nigra* alone or in combination with CPT induced hyperactivation of ATM/ATR-mediated DDR as indicated by elevated level of γH2A histone family member X (γH2AX) as well as increased phosphorylation of ATM, checkpoint kinase 1 (Chk1) and checkpoint kinase 2 (Chk2) when compared to CPT treatment alone (Figure 3A–D). Moreover, overexpression of replication stress-associated DNA repair proteins, including monoubiquitinated FANCD2, FANCJ, and Rad18, were detected following *B. nigra* extract treatment compared to CPT exposure alone. Additionally, the extract alone or in combination with CPT enhanced the expression of cyclin B1 and Bid when compared to control (Figure 3A–D).

### 2.4. B. nigra Extract Altered Cell Cycle Distribution 

The cell cycle analysis indicates that both A549 (Figure 4A,B) and H1299 (Figure 4C,D) cells were significantly delayed and arrested at S and G2/M transition after 24 h treatment with *B. nigra* extract in a concentration-dependent manner.

### 2.5. B. nigra extract Inhibited Migration and Invasion of A549 cells

Migration was assessed after treating A549 cells with different concentrations of *B. nigra* extract (5, 10, and 30 μg/mL) by wound-healing technique. The wound healing technique was performed to estimate the percentage of wound closure after 15 h of *B. nigra* treatment. As shown in Figure 5A,B, *B. nigra* extract significantly suppressed the migration and reduced the wound closure of A549 cells after 15 h in a concentration-dependent manner. After 15 h, the percentage of migratory area covered was 100% for control, while it decreased for the cells treated with *B. nigra* extract in a concentration-dependent manner. This result indicates that the percentage of migratory area of the *B. nigra*-treated cells was significantly less than control, confirming the suppressive effect of *B. nigra* on A549 cell migration. The invasion assay measures the movement of cells through Matrigel, which makes it more similar to what occurs in vivo. Therefore, the trans-well invasion assay was assessed on both A549 and H1299 cells. The results showed that *B. nigra* extract affected the invasive abilities of cells in a concentration-dependent manner. Treatment of A549 cells with *B. nigra* extract at 5, 10, 30, for 15 h (Figure 6A,B) and H1299 cells at 5, 10, and 20 μg/mL for 15 h (Figure 6C,D) significantly suppressed the invasive ability of cancer cells when compared to corresponding control.

### 2.6. B. nigra Altered the Expression of Epithelial-to-Mesenchymal Transition (EMT)-Associated Genes

Regarding the expression of EMT-related molecules, we performed reverse transcription-quantitative polymerase chain reaction (RT-qPCR) for matrix metalloproteinase-2 (MMP2), MMP9, E-cadherin and Snail. Moreover, Western blot was performed to estimate the protein expression of Snail and E-cadherin. Further, the gelatin zymography assay was performed to estimate the expression levels of MMP2 and MMP9 in A549 cells exposed to *B. nigra* extract for 15 h. The cancer metastasis is characterized by the depletion of E-cadherin and upregulation of Snail and metalloproteinases (MMP2 and MMP9) [33]. The qPCR confirmed that *B. nigra*-treated cells exhibited decreased mRNA levels of MMP2, MMP9 and Snail, and increased mRNA level of E-cadherin in a concentration-dependent manner when compared to control for both A549 (Figure 7A,B) and H1299 (Figure 7F,G) cells. Additionally, *B. nigra* extract downregulated the protein expression of Snail and upregulated E-cadherin protein expression in a concentration-dependent manner for A549 (Figure 7C,E) and H1299 cells (Figure 7H,I). The gelatin-zymography revealed the downregulation of MMP2 and MMP9 in A549 cells following the treatment with *B. nigra* extract in a concentration-dependent manner (Figure 7D).

## 3. Discussion

Glucosinolates, the most abundant constituents of cruciferous vegetables (Brassicales), are enzymatically hydrolyzed to release several bioactive components, such as isothiocyanates that act as antitumorigenic agents [34,35]. Brassica vegetables are valuable to humans as they have been associated with cancer chemoprevention [36,37]. Several studies have evaluated the potential health benefits of *B. nigra* as anticonvulsant [38], antioxidant [39], hypoglycemic [40], anticancer [41], and antimicrobial agents [42]. The mechanism of antitumor effect of *B. nigra* seeds is not fully understood, and specific pathways have been previously investigated using limited number of purified components. The previous studies have demonstrated that the chemotherapeutic effects are restricted on G2/M cell cycle arrest, antioxidant effect and mitochondria-mediated apoptosis [27,43,44]. Therefore, the aim of our study was to explore whether *B. nigra* seeds extract affect the proliferation, migration and invasion of A549 and H1299 lung cancer cells. Moreover, we investigated the effect of *B. nigra* treatment on DNA damage and associated repair pathway. In our study, *B. nigra* seeds extract showed inhibition of proliferation of A549 and H1299 cells in a concentration-dependent manner. Moreover, the extract reduced the clonogenic survival of both A549 and H1299 cells. Taken together, our results are in line with a previous study that reported growth-inhibitory effect of mustard seeds constituent allyl isothiocyanate against lung cancer cells [24].

Inhibition of A549 and H1299 cell proliferation by *B. nigra* extract as observed in this study was associated with apoptosis induction and cell cycle arrest in a concentration-dependent manner. *B. nigra*-treated cells showed a 2-fold increase in caspase-3 activity and upregulation of the cleaved caspase-3 compared to control. Nearly 30% and 40% of A549 and H1299 cells were arrested at S and G2/M phases after 30 and 20 μg/mL *B. nigra* treatment when compared with control cells, respectively. Similar results were obtained with allyl isothiocyanate derived from mustard oil [24]. Collectively, these results suggest that *B. nigra* extract exerts antiproliferative effect through the activation of apoptosis and induction of cell cycle arrest. 

Our findings also suggest that *B. nigra* may resemble topo-isomerase enzyme inhibitor CPT in inducing replication-associated DNA damage as demonstrated by upregulation of γH2AX and FANCD2. Moreover, combination of *B. nigra* and CPT enhanced the expression of the replication-associated DNA damage related molecules γH2AX and FANCD2. Accordingly, exposure of A549 and H1299 cells to either *B. nigra* extract alone or in combination with CPT mitigated the progression of cells and accumulated them at the S and G2/M phases. Uncontrolled proliferation of cancer cells attribute to cell cycle deregulation with subsequent dysregulation of cell cycle-related proteins. Further, cyclin B1 plays a prominent role in regulation of cell cycle progression, particularly in G2/M transition phase. Our study revealed that *B. nigra* extract altered the G2/M checkpoint regulator cyclin B1 and increased the G2/M cell cycle arrest in lung cancer cells. Accordingly, the G2/M cell cycle arrest caused by *B. nigra* extract is likely to be associated with upregulation of cyclin B1. During the cell cycle events, particularly at late prophase, cyclin B1-CDK complexes translocate from the cytosol to the nucleus to initiate mitosis. Hence, our data showing upregulation of cyclin B1 in cells after treatment with *B. nigra* extract suggest the failure of nuclear translocation of cyclin B1-CDK complexes. Therefore, upregulation of cyclin B1 after *B. nigra* treatment could result in arresting the cells at G2/M phase and thereby blocking their entry into mitotic events. Importantly, upregulation of DNA damage response signaling within 24 h of *B. nigra* extract exposure indicates that the extract could serve as an effective DNA damaging agent by inducing double strand breaks.

Based on earlier reports, it is likely that cytotoxic activities of different parts of *B. nigra* plant or species of Brassicaceae family are highly variable according to the studied cancer models or various cell lines within the same cancer type [27,30]. In the current study, an extract of *B. nigra* seeds induced replication-associated DNA damage and repair in addition to growth-inhibitory properties as evidenced from the inhibitory effect on the colony formation. DNA repair pathway-associated protein FANCD2 plays a significant role in the stability of genome integrity that is associated with the replication machinery for maintenance of the stability of collapsed replication forks via enhancing the repair of stalled forks by homologous recombination [45]. Previous studies have shown that CPT can induce DNA replication stress and activate Fanconi anemia pathway, resulting in FANCD2 monoubiquitination [32]. In our study, *B. nigra* extract-exposed cells are likely to activate the Fanconi anemia pathway and mediate apoptosis and cell cycle arrest. The *B. nigra*-mediated apoptosis was evidenced by an elevation of caspase-3 activity and upregulation of cleaved caspase-3. Therefore, *B. nigra* can induce potential lethal S-phase-specific DNA lesions. Taken together, these results support that *B. nigra* induced cytotoxic effects similar to the lethal effect of CPT at the S-phase cell cycle [46].

Tumor metastasis is a detrimental consequence of tumor burden; therefore, extensive efforts have been made to identify novel antimetastatic agents [47]. Several studies indicated that cell migration and invasion are involved in tumor metastasis [48,49,50]. Further, the invasiveness of ovarian cancer cells mediated by sirtuin-6 was associated with the expression of EMT-related molecules, such as Snail, E-cadherin, and activated β-catenin [49]. Our findings revealed that *B. nigra* extract inhibited the EMT through suppression of cell migration and invasion when compared to the control cells. The potential inhibition of migration and invasion by *B. nigra* extract was also evidenced by the downregulation of MMP2, MMP9 and Snail genes, and upregulation of E-cadherin gene (hallmark of cancer metastasis) in a concentration-dependent manner at the transcriptional and translational levels. Collectively, our results are consistent with previous report on EMT-inducing effect of sulforaphane (a component of cruciferous vegetables) in bladder cancer cells [48].

The exact phytocomponents of mustard seed (*B. nigra*) extract responsible for the observed effects are not currently known and require additional in-depth investigation. Mustard seeds are known to contain various bioactive phytochemicals. One of the active ingredients of *B. nigra* that showed potential effect against tumors are the isothiocyanates, which are generated by hydrolysis of their precursor compounds glucosinolates. Two important isothiocyanates derived from cruciferous species are sulforaphane and indole-3-carbinol. Sulforaphane has been shown to exert antitumor effects in vitro and in vivo through modulation of various cell signaling pathways [51,52]. Similarly, antitumor activity of indole-3-carbinol has been credited to its ability to interfere with multiple oncogenic signaling pathways controlling cell cycle progression, survival, invasion, and other aggressive behaviors of neoplastic cells [53,54]. Allyl isothiocyanate, an important constituent of *B. nigra*, exhibits anticancer activities in various cancer models, including human lung cancer [24,55]. Hence, it is tempting to speculate that mustard seed phytochemicals may confer the observed antiproliferative, proapoptotic, antimigratory, and anti-invasive activities through synergistic effect.

## 4. Materials and Methods 

### 4.1. Cell Lines and Antibodies

A549 and H1299 cells (classified as NSCLC cells) were collected from American Type Culture Collection (ATCC, Manassas, VA, USA). The cells were cultured in Dulbecco’s modified Eagle’s medium enriched with 10% fetal bovine serum and 100 U/mL penicillin and 100 μg/mL streptomycin sulfate. The possible mycoplasma contamination was checked by Mycotest kit (Invitrogen, Chicago, IL, USA). The following antibodies were used in the study: pATM, FANCD2, Chk1, β-Actin, and glyceraldehyde 3-phosphate dehydrogenase (GAPDH) were obtained from Santa Cruz Biotechnology, Inc. (Santa Cruz, CA, USA); Rad18 from Bethyl Laboratories, Inc. (Montgomery, TX, USA); pATM-Ser1981, pChk1-Ser317, pChk2, FANCJ, Bid, caspase-3, Snail and cyclin B are from Cell Signaling Technologies (Danvers, MA, USA); and γH2AX from Millipore company (Darmstadt, Germany); and E-cadherin from Becton Dickinson company (BD Biosciences, San Jose, CA, USA) Cell Signaling Technologies (The horseradish peroxidase (HRP)-conjugated secondary antibodies, such as anti-mouse IgG-Cy3 and anti-rabbit IgG, were procured from Molecular Probes (Santa Cruz, CA, USA).

### 4.2. Collection and Authentication of Plant Material

The dried *B. nigra* seeds (black mustard) were collected from a local public market. It was identified by Prof. Dr. Ola Hammouda, a botanist in Botany and Microbiology Department, Faculty of Science, Beni-Suef University, Beni-Suef, Egypt.

### 4.3. Preparation of B. nigra Ethanolic Extract

Under sterile conditions, the dried and finely ground seeds of *B. nigra* were subjected to extraction. High grade of absolute ethyl alcohol (≥99.5%, Merck, Darmstadt, Germany) and deionized water were used for the reflux extraction process [56]. The reflux system process was utilized to get crude extract from dry grounded seeds. The process of extraction was performed in a small scale and large-scale reactor (100 mL and 10 L, respectively) by using aqueous ethyl alcohol (50% *v*/*v*). The ratio of seed powder to ethanol was varied from 1:100 and 1:200 at temperature ranges from 60 to 80 °C for 1–3 h. Finally, the seed residues and debris were filtered, and the supernatant was collected. Finally, under vacuum conditions, the collected supernatant was dried by the rotary evaporator system to yield the crude extract in a semi-dried form. 

### 4.4. Cell Viability and Cytotoxicity Assay

A549 and H1299 cells were counted using a Neubauer hemocytometer and nearly 3 × 10^3^ cells were grown in 96-well plate overnight for attachment. After 24 h attachment, the cells were exposed to various concentrations of *B. nigra* extract for 72 h and the control cells were treated by dimethyl sulfoxide (DMSO) to examine the viability and proliferation based on reduction of (3-(4,5-dimethyl thiazolyl-2)-2,5-diphenyltetrazolium bromide (MTT) (Sigma–Aldrich, Saint Louis, MO, USA). After 72 h incubation, cells were washed, trypsinized, and then counted using a Neubauer hemocytometer in presence of trypan blue solution. The cytotoxicity assay was performed using MTT assay. Approximately, 3 × 10^3^ cells were plated and grown in 96-well plate with overnight incubation in humidified conditions of 37 °C and 5% CO_2_. After treatment with various concentrations of *B. nigra* extract (10–80 μg/mL) for 72 h, the media was replaced with MTT (5 mg/mL) solution containing fresh media per well and incubated at 37 °C with 5% CO_2_. After 2-h incubation, the media was decanted, and the formazan crystals were dissolved in 100 μL DMSO. The absorbance of the wells was measured at 560 nm using a Multi-Mode Microplate Reader (Bio-Rad, San Diego, CA, USA). 

### 4.5. Clonogenic Survival Assays

This standard assay is based on the ability of cancer cells to grow and differentiate into colonies in response to drug stimulation. It was performed to determine whether an ethanolic extract of *B. nigra* influences the anchorage-independent growth. A549 and H1299 cells (5 × 10^2^) were seeded in 6-well culture plates overnight and treated with the various concentrations of *B. nigra* extract for 24 h. The cells were then allowed to colonize by replacing the growth medium every 3 days. After 15 days, the colonies were fixed in methanol, followed by staining with crystal violet and subsequently counted using clono-counter software following published method [57].

### 4.6. Apoptosis Assay

The assay was performed as previously described [58]. Caspase-3 activity was measured after treating the cells with DMSO as a control or different concentrations of *B. nigra* extract for 24 h in 96-well plates. After washing twice, the cells were lysed and centrifuged, and the supernatant was then treated with the substrate solution containing benzuloxycarbonyl-ASP-Glu-Val-Asp-aminofluoromethylcoumarin for 1, 2, and 3 h incubation at 37 °C. The fluorogenic substrate for activated caspase-3 (DEVD-AFC) was measured at 1, 2, and 3 h incubation to assess the apoptotic activity after protein estimation using Bradford assay. The free fluorescent due to the generation of 7-amino-4-trifluoromethylcoumarin was measured kinetically after 1, 2, and 3 h incubation as previously described. A slightly modified extraction buffer, containing 25 mM HEPES (pH 7.5), 5 mM MgCl_2_, 1 mM EGTA, pepstatin, leupeptin, and aprotinin (1 μg/mL each), was used for this assay. In addition to the kinetic activity of caspase-3, the cleaved form of caspase-3 (indicator of apoptosis) was evaluated by Western blot after treating the cells with the indicated concentrations of *B. nigra* extract. 

### 4.7. Western Blot

Following treatment of A549 and H1299 cells with *B. nigra* extract and/or CPT at the desired concentrations for 24 h, total proteins were extracted after washing with phosphate-buffered saline (PBS). The cells were lysed in ice-cold cytoskeletal lysis buffer supplemented with freshly prepared phosphatase and protease inhibitors (Roche, Mannheim, Germany). After total protein normalization, samples (50 μg protein) were loaded onto 4x sodium dodecyl sulfate-polyacrylamide gel electrophoresis (SDS-PAGE) gels and separated electrophoretically, and the denatured proteins were transferred to polyvinylidene difluoride nitrocellulose membranes (Millipore, Darmstadt, Germany). At room temperature, the membranes were blocked in 5% dried non-fat milk for 2 h. After blocking, the membranes were incubated with primary antibodies overnight at 4 °C and followed by incubation with HRP-conjugated secondary antibodies [goat anti-rabbit IgG, 1:1000 or anti-mouse IgG, 1:1000] for 2 h at room temperature. The membrane blots were developed using chemiluminescence detection solution. GAPDH and β-actin were used as housekeeping controls. 

### 4.8. Cell Cycle Analysis

Cell cycle analysis was performed as previously described [24]. The cells were incubated in 6-well plates and exposed to DMSO as a control or various concentrations of *B. nigra* extract for 24 h. The cells were harvested after washing two times with PBS, centrifuged and fixed with ice cold 70% ethyl alcohol for 24 h at 4 °C. Subsequently, the cells were stained with propidium iodide solution in the presence of RNAase inhibitor, and finally analyzed by flow cytometry (FACS Calibur, Becton Dickson, San Diego, CA, USA). 

### 4.9. Migration Wound Healing Assay

One of the simplest methods to measure cell migration in various epithelial cell types is wound healing assay [59]. In the current study, the scratch marks were introduced in A549 cells with 100% confluency. After washing with PBS, the cells were exposed to various concentrations (0–30 μg/mL) of *B. nigra* in serum-containing media and allowed to migrate for 15 h. After washing the cells, the wound closure was imaged by a microscope (Nikon, Tokyo, Japan) and estimated using the ImageJ software 1.46r (National Institutes of Health, Bethesda, MD, USA).

### 4.10. Transwell Invasion Assay 

A549 and H1299 cells (1 × 10^5^ cells/well) were seeded in the upper transwell chamber after 15 h exposure to different concentrations of *B. nigra* in 300 μL serum-free medium. The lower chambers were filled with 500 μL 20% FBS-containing medium as chemoattractant. After 15 h incubation, the chambers were washed, fixed using methanol, and finally stained with crystal violet 0.1% (*w*/*v*). The non-invaded cells were removed with cotton swab from the upper chamber and 5 fields from the outer chamber were counted for the number of invaded cells using a light microscope [50].

### 4.11. In Situ Gelatinase Zymography 

Gelatinase zymography is a relatively simple technique which utilizes enzymatic substrate-based overlay to disclose gelatinases activity and detect both pro and active forms of MMP2 and MMP9. The gelatinase hydrolyzes the gelatin in the gel and reveals the clear bands relative to metalloproteinases. The zymographic gel was freshly prepared in 8% resolving SDS-PAGE in presence of 0.1% gelatin type B with non-reducing conditions. After exposing A549 cells to various concentrations (5, 10, and 30 μg/mL) of *B. nigra* extract for 15 h, the supernatant was collected and preserved at −20 °C until used. For the assay, 20 μL of the cultured media were mixed with sample buffer in room temperature for 10 min without heating before loaded onto SDS-PAGE provided with tris-glycine SDS buffer. Following subsequent electrophoresis, the gels were removed carefully and washed thrice with 2.5% Triton X-100 washing buffer for 15 min each at room temperature on a rotary shaker. Then the gel was incubated at 37 °C with zymographic development buffer (50 mM Tris-HCl and 10 mM CaCl_2_ at pH 7.4) for 42 h. The gel was then stained with 0.125% Commassie Blue R-250 in 45% ethanol and 1% glacial acetic acid for 1 h. After 3 h of gel destaining, the proteins were clearly disclosed approximate to their molecular weights and the gelatinase zymograms were scanned using Las-3000 luminescent image analyzer (Fuji Film, Tokyo, Japan) [60].

### 4.12. Gene Expression Studies 

After 24 h of *B. nigra* treatment, total RNA was isolated with TRIzol RNA lysis buffer (Qiagen, Hilden, Germany). The purity and concentration of RNA were determined using NanoDrop™ 2000/2000c Spectrophotometer (ThermoFisher, Waltham, MA, USA) followed by DNA digestion. The cDNA synthesis was obtained by reverse transcription of 1 μg RNA using RT-qPCR kit (Takara Biotechnology Co. Ltd., Dalian, China) based on the manufacturer’s instructions. The used gene primers are provided in Table 1. The real-time PCR was set based on the following conditions: 5 μL 2X SYBR-Green mixture (Qiagen GmbH, Hilden, Germany), 1 μL primer and 4 μL double-distilled water. After denaturation at 95 °C, 45 PCR cycles were performed using ViiA7 quantitative fluorescence PCR machine (ABI Corporation, Lee’s Summit, MO, USA) at 95 °C for 10 sec, 55 °C for 40 sec, and 72 °C for 25 sec. The samples were set in triplicates and the genes were normalized to GAPDH as a housekeeping gene.

### 4.13. Statistical Analysis

To confirm the reproducibility of the results, the experiments were performed in triplicate. The data, expressed as mean ± standard error of means (SEM), were analyzed using GraphPad prism V.6.0.0 software (GraphPad, San Diego, CA, USA).

## 5. Conclusions

Based on the data presented in this work, we conclude that an ethanolic extract of *B. nigra* seeds exhibits substantial antiproliferative activity in A549 and H1299 human lung cancer cells possibly by induction of apoptosis and regulation of cell cycle via replication stress, resulting in fork-collapse DNA lesions. Moreover, our results indicate antimetastatic potential of *B. nigra* extract by the virtue of its antimigratory and anti-invasive activities possibly through downregulation of MMP2, MMP9, Snail and upregulation of E-cadherin (Figure 8). Overall, this study suggests that *B. nigra*-derived phytoconstituents could be developed as a potential chemopreventive and therapeutic agent for lung cancer. However, additional studies, including in vivo experiments, are needed to realize the full impact of the results reported here.

## Figures and Tables

**Figure 1 molecules-25-02069-f001:**
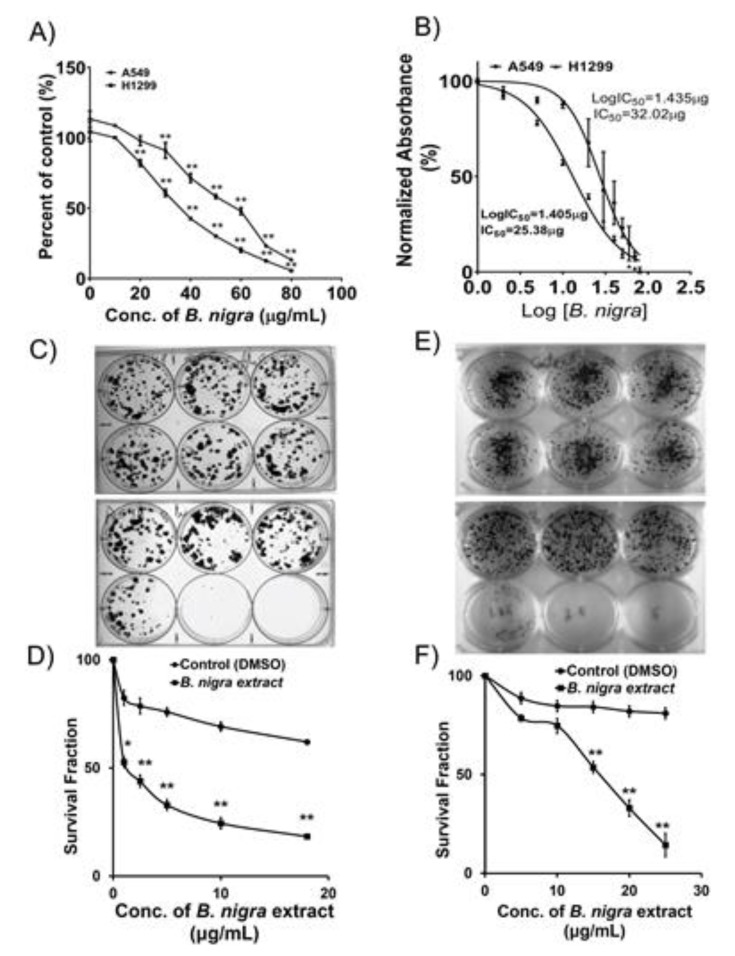
*B. nigra* extract inhibited proliferation and clonogenic survival of A549 and H1299 cells. (**A**) A549 and H1299 cells were treated with the indicated concentrations of *B. nigra* extract for 72 h and the viability of cells was examined with trypan blue solution and proliferation was measured using MTT assay. The results show that *B. nigra* extract inhibited the proliferation of both A549 and H1299 in a concentration-dependent manner. (**B**) The cytotoxic effect of *B. nigra* extract on A549 and H1299 cells was evidenced based on IC_50_ values of 32.02 and 25.38 μg/mL, respectively. (**C**) A549 cells treated with various concentration of dimethyl sulfoxide (DMSO) (upper panel) or *B. nigra* extract (lower panel) for 24 h for clonogenic assay. (**E**) H1299 cells treated with DMSO (upper panel) or *B. nigra* extract (lower panel). (**D**,**F**) *B. nigra* extract inhibited the formation of colony of A549 (**D**) and H1299 (**F**) cells in a concentration-dependent manner. All experiments were performed in triplicates for each concentration and data are presented as mean ± SEM. * *p* < 0.05 and ** *p* < 0.01 compared to control.

**Figure 2 molecules-25-02069-f002:**
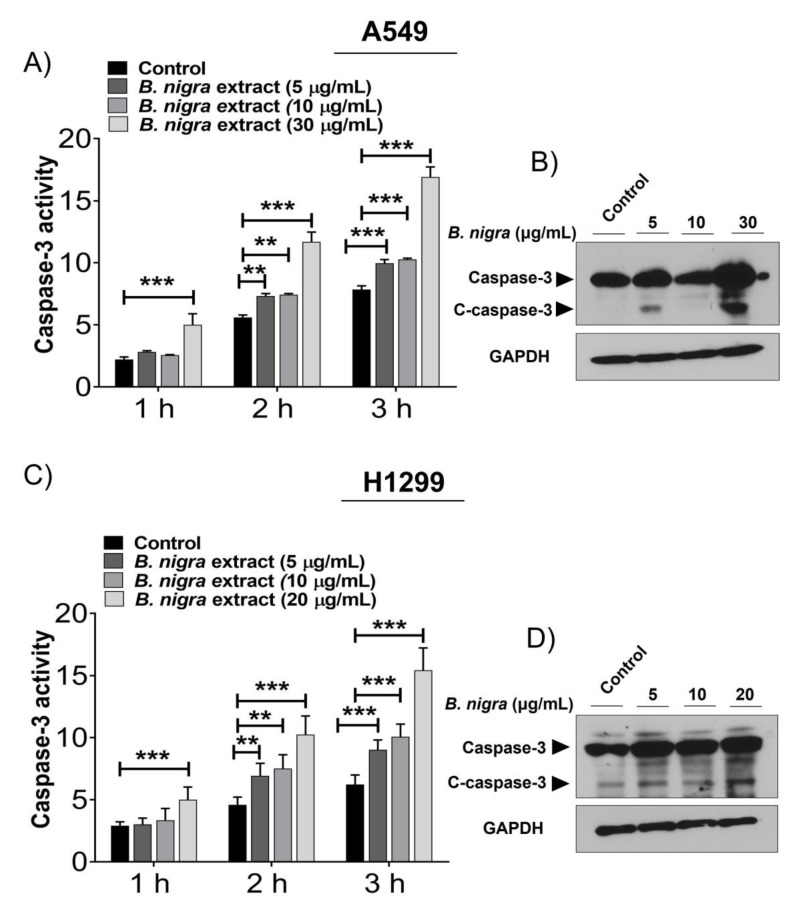
*B. nigra* extract induced apoptosis in A549 and H1299 lung cancer cells. Both A549 and H1299 cells were treated with *B. nigra* extract at various indicated concentrations for 24 h. The extent of apoptosis was measured based on the kinetic activity of caspase-3 after 1, 2 and 3 h incubation with the substrate solution containing ASP-GLu (**A**,**C**) and the protein expression of cleaved caspase-3 (**B**,**D**) of A549 and H1299 cells. The data was generated from 3 independent experiments and presented as mean ± SEM. ** *p* < 0.05 and *** *p* < 0.001 compared to control.

**Figure 3 molecules-25-02069-f003:**
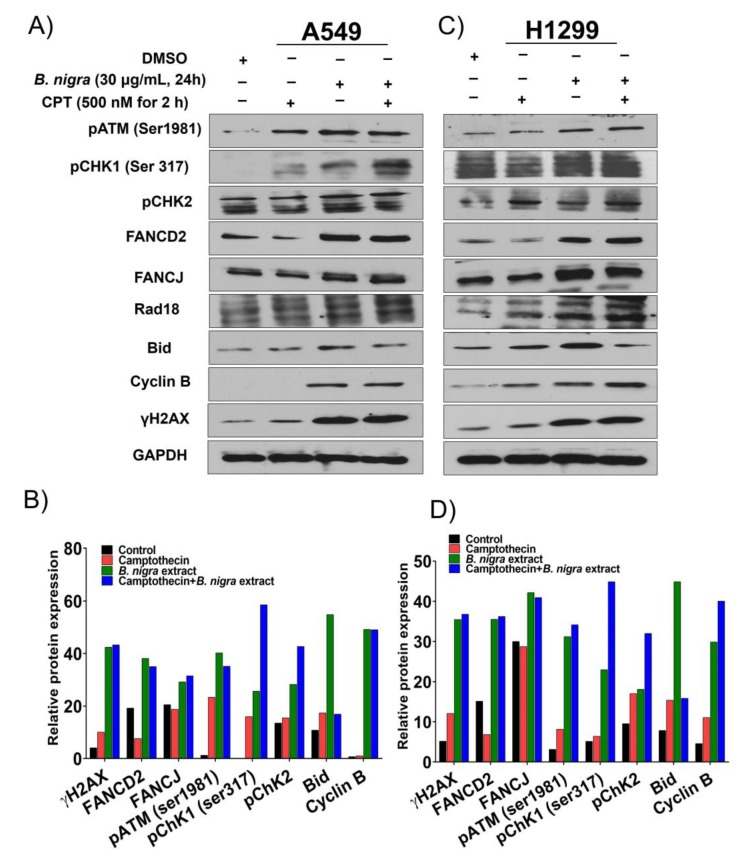
*B. nigra* extract induced replication-associated DNA lesion and stimulated cell cycle checkpoint response in A549 and H1299 lung cancer cells. (**A**,**C**) The A549 and H1299 cells were treated with *B. nigra* extract at a concentration 30 and 20 μg/mL for 24 h, respectively. The culture medium and the whole protein was then collected, normalized, resolved on SDS-PAGE and probed with antibodies for different DNA damage and repair proteins. (**B**,**D**) Quantitative analysis of protein expressions are shown.

**Figure 4 molecules-25-02069-f004:**
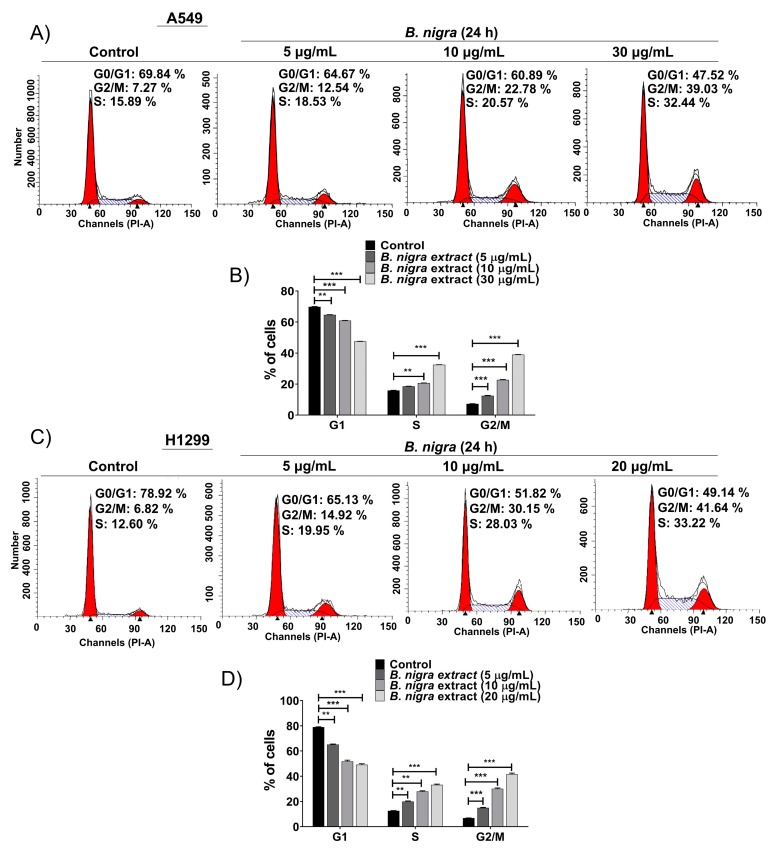
*B. nigra* extract induced accumulation of A549 and H1299 lung cancer cells at S and G2/M phases. (**A**,**C**) Cell cycle distribution of A549 and H1299 cells exposed to *B. nigra* at different concentrations for 24 h was analyzed by flow cytometry, respectively. (**B**,**D**) The percentage of cell population at each cell cycle phase is depicted. The data generated from 3 independent experiments were presented as mean ± SEM. ** *p* < 0.01 and *** *p* < 0.001 compared to control.

**Figure 5 molecules-25-02069-f005:**
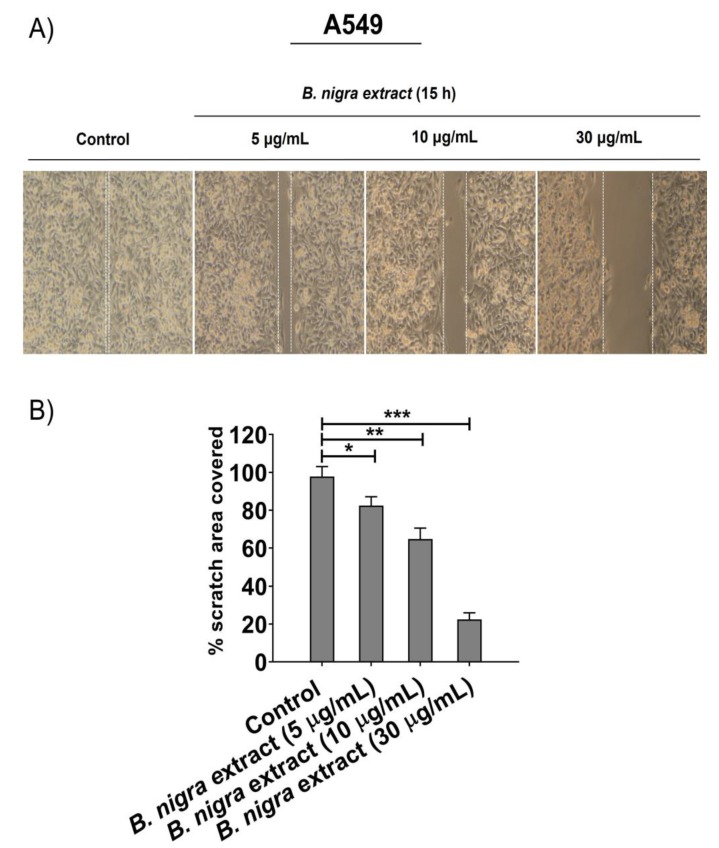
*B. nigra* extract is a potent suppressor of A549 lung cancer cell migration. (**A**). Cell migration was measured by scratch-wound healing assay after incubation of A549 cells with the indicated concentrations of *B. nigra* extract for 15 h. The representative images of wound were taken before and after *B. nigra* extract treatment. (**B**) The percentage of scratch area closure * *p* < 0.5, ** *p* < 0.01 and *** *p* < 0.001 were significant compared to control.

**Figure 6 molecules-25-02069-f006:**
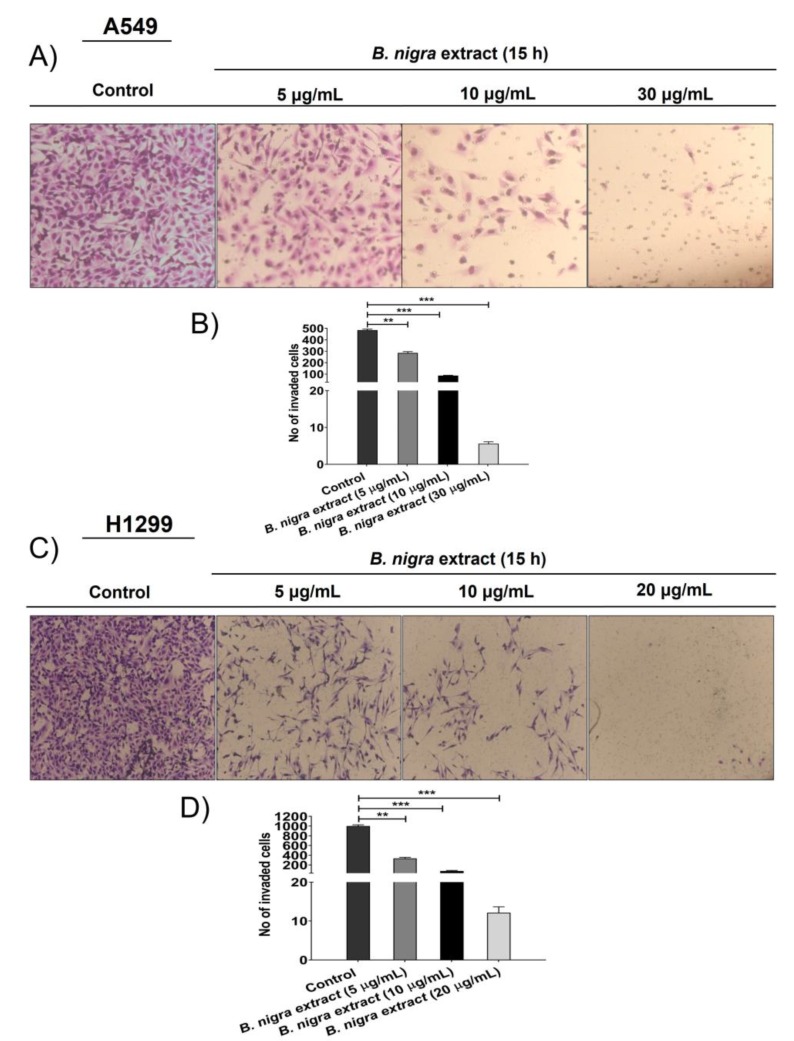
*B. nigra* extract is a potent suppressor of A549 and H1299 lung cancer cell invasion. (**A**,**C**) Cell invasion was estimated by trans-well invasion assay after exposing A549 and H1299 cells to different concentrations of *B. nigra* extract for 15 h. The chambers were stained, after removing the non-invaded cells from the inner chamber, by crystal violet and then imaged for estimating the number of invaded cells. (**B**,**D**) The number of invaded cells were counted and quantified based on triplicate experiments and the mean values were represented by mean ± SEM ** *p* < 0.01 and *** *p* < 0.001 compared to control.

**Figure 7 molecules-25-02069-f007:**
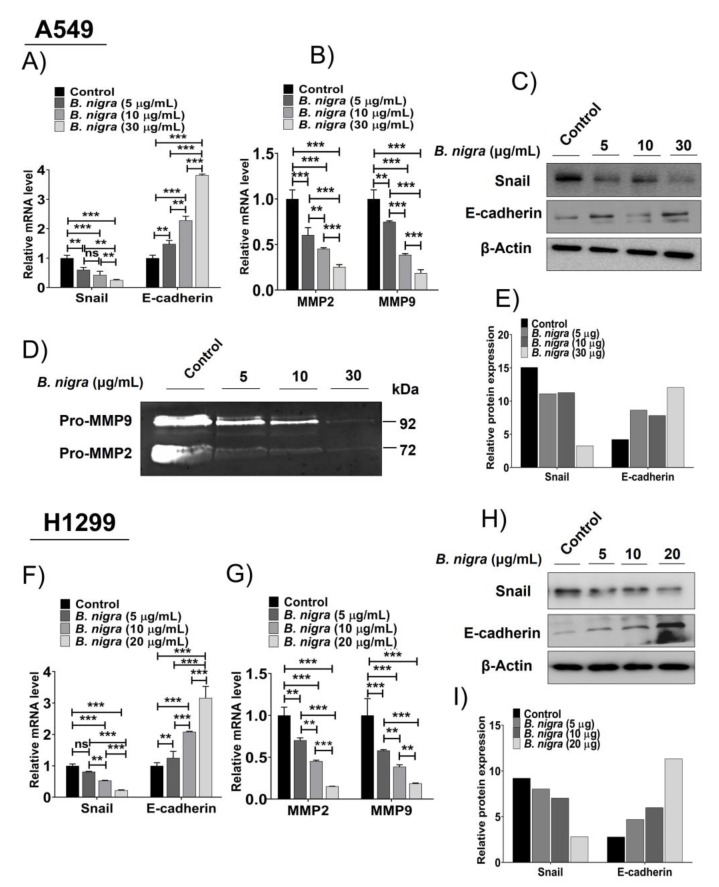
*B. nigra* extract modulated epithelial-to-mesenchymal transition (EMT)-associated gene expression in A549 and H1299 lung cancer cells. A549 and H1299 cells were treated with the indicated concentrations of *B. nigra* extract for 24 h, and then qRT-PCR (**A**, **B**, **F**, and **G**) and Western blot (**C**, **E**, **H**, and **I**) assays were performed to determine the expression of Snail, E-cadherin, MMP2, and MMP9 at transcriptional and translational levels, respectively. Moreover, the gelatin zymography assay was performed for A549 cells exposed to *B. nigra* extract at the indicated concentrations for 15 h to detect the degree of MMP2 and MMP9 expression levels (**D**). ** *p* < 0.01 and *** *p* < 0.001 were significant compared to control.

**Figure 8 molecules-25-02069-f008:**
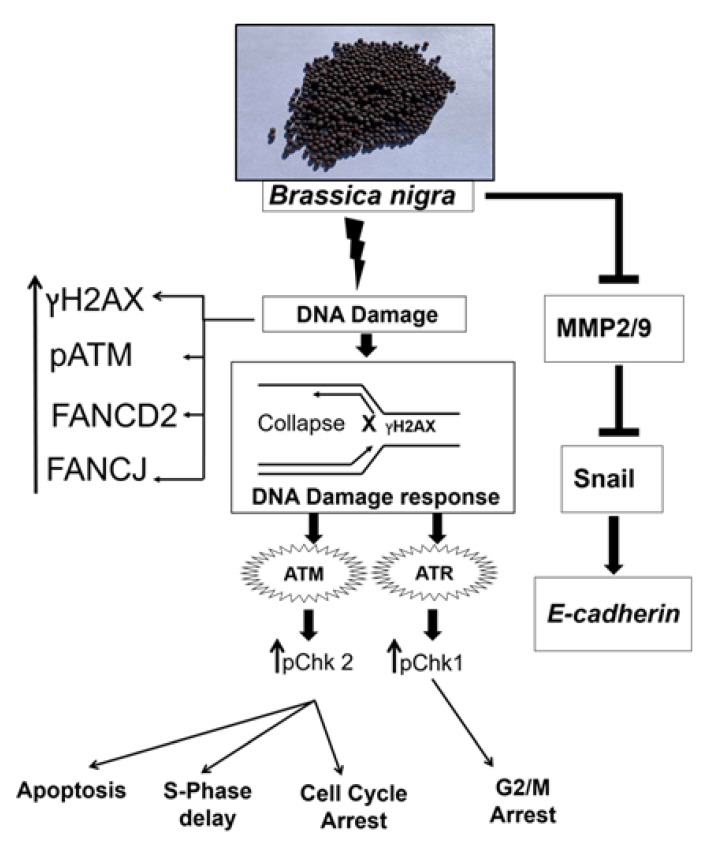
Overview of anticancer mechanisms of *B. nigra* seeds extract in human lung cancer cells. *B. nigra* extract induced replication-associated DNA damage response followed by stimulation of cell cycle checkpoints and apoptosis. Additionally, *B. nigra* extract inhibited the metalloproteinase enzymes, MMP2 and MMP9, which in turn inhibited downstream Snail and induced E-cadherin expression.

**Table 1 molecules-25-02069-t001:** Primer sequences used for reverse transcription-quantitative polymerase chain reaction (RT-qPCR).

Gene	Primer Sequence		Product Size	Accession Number
*E-cadherin*	Forward	TGGGCCAGGAAATCACATCC	863	NM_001317185.2
	Reverse	TGCAACGTCGTTACGAGTCA		
*SNAL1* (Snail)	Forward	CGAGTGGTTCTTCTGCGCTA	160	NM_005985.3
	Reverse	GGGCTGCTGGAAGGTAAACT		
*MMP2*	Forward	CGGCCGCAGTGACGGAA	212	NM_004530.4
	Reverse	CATCCTGGGACAGACGGAAGTTCTT		
*MMP9*	Forward	GACGCAGACATCGTCATCCA	200	NM_004994.2
	Reverse	GCCGCGCCATCTGCGTTTCCAAA		
*GAPDH*	Forward	AACAGCGACACCCACTCCTC	258	NM_001256799.1
	Reverse	GGAGGGGAGATTCAGTGTGGT		

Web link to accession numbers: https://www.ncbi.nlm.nih.gov/gene.

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
