# Peer review of "Mustard Seed (Brassica nigra) Extract Exhibits Antiproliferative Effect against Human Lung Cancer Cells through Differential Regulation of Apoptosis, Cell Cycle, Migration, and Invasion"

_molecules, 2020, doi:10.3390/molecules25092069_

Round 1

Reviewer 1 Report

In this study, the authors examined the anti-effects of Mustard seed (B. nigra) extract on human lung cancer cell, A549. The authors suggested that in A549, there are two anti-cancer pathways, one in which B. nigra extract causes direct DNA damage and subsequent apoptosis and cell arrest, and another in which invasion is suppressed through direct inhibition of MMPs. The results obtained from the B. nigra extract effect are interesting, but they are very limited.

The results that were elucidated using A549 do not apply to all or certain lung cancer cells. The problem is that the study was limited to just A549 alone. It is well recognized that A549 has wildtype EGFR and KRAS mutation. Why did you use A549 for this study? In order to obtain more clinically significant in vitro results, it is necessary to examine using lung cancer cell lines of several characteristics.

In this study, the authors examined the change in e-cadherin expression levels induced by B. nigra using quantitative PCR. In some reports, A549 naturally expresses e-cadherin, the change of which uses protein. In some reports, A549 naturally expresses e-cadherin, and their differences have been evaluated using proteins and specific its antibody. Each factor including e-cadherin should be examined at the protein level.

In Fig. 7, the low concentration B. nigra extract markedly increased the expression of Snail and MMPs as compared to the control. Otherwise, when the high concentration B. nigra extract was added, their expression levels were not significantly affected compared to the control. The interpretation of these findings should be clear in the discussion section.

High concentrations of B. nigra extract significantly blocked A539 cell migration/invasion. However, it is also true that this high concentration extract caused cellular and DNA damage to many cells. Under these conditions, the migrative ability of A549 will be strongly affected.

Author Response

The authors of this manuscript express their sincere thanks to the reviewer for the critical assessment of our work. The authors have acted upon the recommendations of the reviewer which have resulted in a significant enhancement of the quality of this manuscript. All modifications incorporated in the manuscript are highlighted using red color font. A “point-by-point” response to the reviewers’ comments is outlined below.

Comment 1:

In this study, the authors examined the anti-effects of Mustard seed (B. nigra) extract on human lung cancer cell, A549. The authors suggested that in A549, there are two anti-cancer pathways, one in which B. nigra extract causes direct DNA damage and subsequent apoptosis and cell arrest, and another in which invasion is suppressed through direct inhibition of MMPs. The results obtained from the B. nigra extract effect are interesting, but they are very limited.

Response:

We thank the reviewer for this excellent comment.

The reviewer has correctly pointed out that we have investigated two pathways: one related to DNA damage response and repair related pathway, and the other related to epithelial-to-mesenchymal transition (EMT) pathway. On the second pathway, we found a potential inhibitory effect of Brassica nigra extract on migration and invasive ability of human lung cancer cells. Our results on the second part also showed inhibition of MMP2, MMP9, and Snail which directly upregulated the E-cadherin expression in a concentration-dependent manner.

We agree with the reviewer’s assessment (and also the next comment) and have performed additional experiments using another lung carcinoma cell line H1299 to get more clinically-relevant results. The new results are incorporated in several figures (Figures 1, 2, 3, 4, 6 and 7) and also described in the appropriate result section (pages 3-9) of the revised manuscript.

Comment 2:

The results that were elucidated using A549 do not apply to all or certain lung cancer cells. The problem is that the study was limited to just A549 alone. It is well recognized that A549 has wildtype EGFR and KRAS mutation. Why did you use A549 for this study? In order to obtain more clinically significant in vitro results, it is necessary to examine using lung cancer cell lines of several characteristics.

Response:

We are indebted to the reviewer for this valuable comment. We are in absolute agreement with the reviewer and performed additional experiments by using another human non-small cell lung carcinoma cell line (H1299) in order to get more clinically significant results. The H1299 cells were originally derived from metastatic lymph nodes of a cancer patient. First, we evaluated the cytotoxicity of B. nigra extract on H1299 cells by MTT assay followed by determination of the survival of cells using colony-forming assay. Moreover, the DNA damage response and repair checkpoints have been detected after treatment of the cells with B. nigra extract with subsequent analysis of the cell cycle for H1299 cells. The results showed delay and arrest in S-phase and G2/M cell cycle transition. Furthermore, the apoptosis was evaluated after B. nigra treatment by using caspase-3 kinetic assay and Western blot analysis, which indicated elevated expression of the cleaved caspase-3 in a concentration-dependent manner. Additionally, we examined the potential inhibitory effect of different concentrations of B. nigra on the invasiveness of H1299 cells. Furthermore,  the mRNA expressions of matrix metalloproteinase-2 (MMP2), MMP9, and Snail, E-cadherin were determined by RT-qPCR and protein expressions of Snail and E-cadherin were investigated by Western blot. All new results are incorporated in the results section - both in figures (Figures 1, 2, 3, 4, 6 and 7) and text (pages 3-9).

Comment 3:

In this study, the authors examined the change in e-cadherin expression levels induced by B. nigra using quantitative PCR. In some reports, A549 naturally expresses e-cadherin, the change of which uses protein. In some reports, A549 naturally expresses e-cadherin, and their differences have been evaluated using proteins and specific its antibody. Each factor including e-cadherin should be examined at the protein level.

Response:

We admire the reviewer for this thought-provoking comment and important recommendation. Accordingly, we evaluated the EMT markers, such as Snail and E-cadherin, by Western blot analysis on both A549 and H1299 human non-small cell lung carcinoma cell lines in addition to the qPCR analysis. We have revised Figure 7 (page 9) to incorporate the Western blot data and the description of results is added to the results section (page 9, lines 205-207).

Comment 4:

In Fig. 7, the low concentration B. nigra extract markedly increased the expression of Snail and MMPs as compared to the control. Otherwise, when the high concentration B. nigra extract was added, their expression levels were not significantly affected compared to the control. The interpretation of these findings should be clear in the discussion section.

Response:

We thank the reviewer for this close observation and comment. We have repeated the qPCR experiment for the A549 cells after treatment with B. nigra extract to ensure the accuracy of results. The qPCR analysis showed that the relative mRNA levels of MMP2, MMP9 and Snail were significantly decreased when compared to control upon treatment with B. nigra extract in a concentration-dependent manner. In addition, we evaluated the qPCR for E-cadherin as well to confirm the previous results. We revised Figure 7 (page 9) accordingly and reported the latest results in the text (page 9, lines 202-205).

Comment 5:

High concentrations of B. nigra extract significantly blocked A549 cell migration/invasion. However, it is also true that this high concentration extract caused cellular and DNA damage to many cells. Under these conditions, the migrative ability of A549 will be strongly affected.

Response:

We thank the reviewer for this acute observation and comment. We performed the cytotoxicity and colony-forming assays by incubation of both A549 and H1299 cells with B. nigra extract for 24 h. The DNA damage was also assessed following 24-h treatment with the extract. The Western blot and qPCR experiments were analyzed after incubating the cells with B. nigra extract for 24 h as well. On the other hand, the migration and invasion assays have been performed after incubating the cells with B. nigra extract for 15 h only. We have used decreased incubation time for invasion and migration assays because a 24-h treatment of lung cancer cells with B. nigra extract has cytotoxic effect on cancer cells by inducing DNA damage and apoptosis. Thus, we decreased the incubation time for the migration and invasion assays to avoid the cytotoxic effects on the migration and invasion ability of the cells. We sincerely apologized for our inadvertent oversight and provided accurate information regarding the treatment time (15 h) for migration and invasion assay in the Materials and Methods section (page 13, line 389). 

On behalf of my co-authors, I once again express my sincere thanks to the erudite reviewers and the academic editor for the valuable suggestions and constructive input to improve the quality of our manuscript.

Reviewer 2 Report

Dear authors,

I found the manuscript interesting and well written. I recommend you to include a chromatographic profile of the extract indicating its major compounds.

Authors  said "Hence, it is tempting to speculate that mustard seed phytochemicals may confer the observed antiproliferative, proapoptotic, antimigratory and anti-invasive activities through synergistic effect"

Which phytochemicals? Why authors think about a synergistic effect? Have you tested any compound alone? 

Author Response

The authors of this manuscript express their sincere thanks to the reviewer for the critical assessment of our work. The authors have acted upon the recommendations of the reviewer which have resulted in a significant enhancement of the quality of this manuscript. All modifications incorporated in the manuscript are highlighted using red color font. A “point-by-point” response to the reviewers’ comments is outlined below.

Comment 1:

I found the manuscript interesting and well written. I recommend you toinclude a chromatographic profile of the extract indicating its major compounds.

Response:

We thank the reviewer for the appreciation of our work. We are unable to provide a chromatographic profile of the extract to indicate its major compounds due to lack of facility and resources for this experiment as well as the availability of the very limited amount of time to submit a revised manuscript. We certainly believe this is an excellent suggestion, and we look forward to investigating phytochemical profiling in the future as indicated in the manuscript (page 11, lines 284 and 285). However, we would like to mention that the phytochemical analysis of Brassica nigra and other Brassicacea family plants have been provided in a wide range of previous reports. In these publications, the investigators showed the presence of various active constituents of plants of Brassica family, including B. nigra, such as glucosinolates which are hydrolyzed by myrosinase enzymes to yield isothiocyanates, including allyl isothiocyanate, phenethyl isothiocyanates, sulforaphane, and phenyl isothiocyanate. We have mentioned the major components of B. nigra seeds in our manuscript (page 2, lines 70-72; page 11, lines 286-294).

Comment 2:

Authors  said "Hence, it is tempting to speculate that mustard seed phytochemicals may confer the observed antiproliferative, proapoptotic, antimigratory and anti-invasive activities through synergistic effect"

Which phytochemicals? Why authors think about a synergistic effect? Have you tested any compound alone?

Response:

We greatly appreciate this thought-provoking comment. Based on studies presented in the manuscript, it is likely that various active phytochemicals of B. nigra may have a synergistic effect in inhibiting the proliferation, migration and invasion and promoting the apoptosis of tumor cells. Emerging evidence suggests that plant phytochemicals exert anticancer effects when they are used in combination rather than individually. Based on previous reports, various plant phytochemicals from dietary sources, including cranberry, raspberry, pomegranate and green tea, may exert better cancer suppressive activities when used in combination rather than in single pure form (Seeram et al., 2004; Lansky et al., 2005; de Kok et al., 2008; Bode and Dong, 2009; Zikri et al., 2009; full references are provided below). Hence, it plausible that B. nigra phytochemicals may confer the observed activities via the promotion of multifactorial effects, utilizing chemical synergy. Obviously, more research is needed to understand the complex phytochemical synergy of B. nigra for a complex disease, such as cancer. We take this comment of the reviewer as a guide and inspiration for performing additional studies in the future as indicated in the manuscript (page 11, lines 284 and 285).

  • Bode AM, Dong Z. Epigallocatechin 3-gallate and green tea catechins: United they work, divided they fail. Cancer Prev Res (Phila). 2009;2:514-7.
  • de Kok TM, van Breda SG, Manson MM. Mechanisms of combined action of different chemopreventive dietary compounds. Eur J Nutr 2008;47:51–9.
  • Lansky EP, Jiang W, Mo H, Bravo L, Froom P, Yu W, et al. Possible synergistic prostate cancer suppression by anatomically discrete pomegranate fractions. Invest New Drugs 2005;23:11–20.
  • Seeram NP, Adams LS, Hardy ML, Heber D. Total cranberry extract versus its phytochemical constituents: antiproliferative and synergistic effects against human tumor cell lines. J Agric Food Chem 2004;52:2512–7.
  • Zikri NN, Riedl KM, Wang LS, Lechner J, Schwartz SJ, Stoner GD. Black raspberry components inhibit proliferation, induce apoptosis, and modulate gene expression in rat esophageal epithelial cells. Nutr Cancer 2009;61:816–26.

On behalf of my co-authors, I once again express my sincere thanks to the erudite reviewers and the academic editor for the valuable suggestions and constructive input to improve the quality of our manuscript.

Reviewer 3 Report

Manuscript ID: molecules-757268

Title: Mustard seed (Brassica nigra) extract exhibits antiproliferative effect against human lung cancer cells through differential regulation of apoptosis, cell cycle, migration, and invasion

  • B. nigra → B. nigra (italic) (overall manuscript)
  • Fig.1C: please present the sample name and control. Which well is a control or sample treatment?
  • Fig.3 A: B. nigra treatment showed inhibited proliferation and cell cycle arrest. But cyclin B is upregulated at B. nigra treatment and B. nigra + CPT. Authors should address this and explain the reason.

Author Response

The authors of this manuscript express their sincere thanks to the reviewer for the critical assessment of our work. The authors have acted upon the recommendations of the reviewer which have resulted in a significant enhancement of the quality of this manuscript. All modifications incorporated in the manuscript are highlighted using red color font. A “point-by-point” response to the reviewers’ comments is outlined below.

Comment 1:

  1. nigra → B. nigra (italic) (overall manuscript)

Response:

We thank the reviewer for this comment. We have corrected it throughout the manuscript.

Comment 2:

Fig.1C: please present the sample name and control. Which well is a control or sample treatment?

Response:

We thank the reviewer for this comment. We sincerely apologize for the inadvertent error and have modified the legend to Figures 1C and 1E to indicate the exact treatment of cell lines used in our study (page 3, lines 113-115).

Comment 3:

Fig.3 A: B. nigra treatment showed inhibited proliferation and cell cycle arrest. But cyclin B is upregulated at B. nigra treatment and B. nigra + CPT. Authors should address this and explain the reason.

Response:

We thank the reviewer for this critical observation. We addressed this point in the results section (page 6, lines 155 and 156) and the interpretation of the results are added to the discussion section (page 10, lines 244-254).

On behalf of my co-authors, I once again express my sincere thanks to the erudite reviewer for the valuable suggestions and constructive input to improve the quality of our manuscript.

Round 2

Reviewer 1 Report

The submitted revision has been significantly revised. However, there are still some unknowns.

Migration and invasion of lung cancers were observed 15 hours after dosing, while EMT genes expressions were examined 24 hours later. Phenotypic changes are thought to occur after genetic changes. The authors should explain the time lag between those examinations. Caspase expression was examined 3 hours after dosing, and proliferation assay were evaluated 72 hours later. The timing of all studies should be mentioned. I think the effect on migration may be due to cell damage

The significance of some extra bands in western blotting should be explained.

It is necessary to describe the characteristics of the two lung adenocarcinoma cell lines employed in this study.

Author Response

The authors of this manuscript express their sincere thanks to the reviewer for the critical assessment of our work. The authors have acted upon the recommendations of the reviewer which have resulted in a significant enhancement of the quality of this manuscript. All modifications incorporated in the manuscript are highlighted using red color font. A “point-by-point” response to the reviewer's comments is outlined below.

Comment 1:

The submitted revision has been significantly revised. However, there are still some unknowns.

Response:

We are grateful to the reviewer for his/her time in evaluating our revised manuscript and also for the appreciation of our effort to make the necessary modifications. We welcome the reviewer’s additional comments and addressed them as described below.

Comment 2:

Migration and invasion of lung cancers were observed 15 hours after dosing, while EMT genes expressions were examined 24 hours later. Phenotypic changes are thought to occur after genetic changes. The authors should explain the time lag between those examinations. Caspase expression was examined 3 hours after dosing, and proliferation assay were evaluated 72 hours later. The timing of all studies should be mentioned. I think the effect on migration may be due to cell damage.

Response:

We thank the reviewer for this thought-provoking comment. In line with previous studies, we have determined the 50% inhibitory concentration (IC50) of Brassica nigra extract following the treatment of A549 and H1299 lung cancer cell lines for 72 h. For migration and invasion assay, although we have used a treatment period of 15 h, the concentration of the extract was less than the IC50 values. We believe that due to shorter exposure time and a lower concentration of the extract, the cells may have experienced a lesser degree of cytotoxic effect for the migration and invasion assays. We have also performed additional experiments and included data to show the downregulation of MMP2 and MMP9 by B. nigra extract 15 h following the treatment. Our new results may indicate that certain molecular changes associated with migration and invasion may occur around the same time as the phenotypical alterations.

For caspase assay, we treated the cells for 24 h and the time-course enzymatic assay was performed at 1, 2, or 3 h following the treatment period. This has been clearly indicated in the manuscript (page 13, Section 4.6).

We have made the necessary changes to clearly indicate the exposure time for various assays (see the table below). The information has now been uniform and consistent for the various parts of the manuscript, e.g., results, materials and methods, figures, and their legends. 

Assays described in the manuscript

B. nigra treatment time (h)

Cell viability and proliferation (Trypan blue and MTT)

72

Cytotoxicity (MTT)

72

Clonogenic survival

24

Apoptosis (caspase-3)

24

Western blot

24

Cell cycle

24

Migration (Wound healing)

15

Invasion (Transwell)

15

Gelatinase zymography

15

RT-qPCR

24

Comment 3:

The significance of some extra bands in western blotting should be explained.

Response:

We greatly appreciate the reviewer’s watchful eyes. With regards to the extra bands which are very close to the molecular weight of the protein of interest, in our experience, it is a very common feature of Western blotting, and we believe everyone who does immunoblotting faces it quite often. Further, for blocking, we used 5% skim milk in TBST or 3% BSA in TBST. We generally keep it for blocking for 1 h. However, 2-h blocking is also used by many researchers. Another important factor of interference could be the antibodies used. Many commercially available antibodies are not very specific and might bind to different proteins. Moreover, the exposure time could play a role in the appearance of non-specific bands. In order to visualize the expression of various phosphorylated proteins, the long exposure time was used to generate data as presented in Figures 2 and 3. The E-cadherin is always expressed as two close bands as depicted in Figure 7. All these features, however, do not alter the differential results in control and B. nigra-treated cells and may have no impact on the overall findings and conclusions of this study.

Comment 4:

It is necessary to describe the characteristics of the two lung adenocarcinoma cell lines employed in this study.

Response:

We thank the reviewer for this recommendation. We have described the characteristics of cells used in this study in the introduction section (page 2, lines 93-95) and cited appropriate reference (ref. no. 31).

On behalf of my co-authors, I once again express my sincere thanks to the erudite reviewer for the valuable suggestions and constructive input to improve the quality of our manuscript.